# *Streptomyces pratensis* S10 Promotes Wheat Plant Growth and Induces Resistance in Wheat Seedlings against *Fusarium graminearum*

**DOI:** 10.3390/jof10080578

**Published:** 2024-08-15

**Authors:** Xiaoman Tian, Lifang Hu, Ruimin Jia, Shang Cao, Yan Sun, Xiaomin Dong, Yang Wang

**Affiliations:** 1College of Bioengineering, Yangling Vocation & Technical College, Yangling, Xianyang 712100, China; txiaom123@163.com; 2College of Plant Protection, Northwest A&F University, Yangling, Xianyang 712100, China; hulifang@nwafu.edu.cn (L.H.); jiaruimin@nwafu.edu.cn (R.J.); caoshang@nwafu.edu.cn (S.C.); sunyan5648@nwafu.edu.cn (Y.S.); dongxiaomin@nwafu.edu.cn (X.D.)

**Keywords:** *Streptomyces pratensis* S10, plant growth promotion, genome mining, callose deposition, defense-related enzymes

## Abstract

*Fusarium graminearum*, a devastating fungal pathogen, causes great economic losses to crop yields worldwide. The present study investigated the potential of *Streptomyces pratensis* S10 to alleviate *F. graminearum* stress in wheat seedlings based on plant growth-promoting and resistance-inducing assays. The bioassays revealed that S10 exhibited multiple plant growth-promoting properties, including the production of siderophores, 1-aminocyclopropane-1-carboxylic acid deaminase (ACC), and indole-3-acetic acid (IAA), phosphate solubilization, and nitrogen fixation. Meanwhile, the pot experiment demonstrated that S10 improved wheat plant development, substantially enhancing wheat height, weight, root activity, and chlorophyll content. Consistently, genome mining identified abundant genes associated with plant growth promotion. S10 induced resistance against *F. graminearum* in wheat seedlings. The disease incidence and disease index reduced by nearly 52% and 65% in S10 pretreated wheat seedlings, respectively, compared with those infected with *F. graminearum* only in the non-contact inoculation assay. Moreover, S10 enhanced callose deposition and reactive oxygen species (ROS) accumulation and induced the activities of CAT, SOD, POD, PAL, and PPO. Furthermore, the quantitative real-time PCR (qRT-PCR) results indicated that S10 pretreatment increased the expression of SA- (*PR1.1*, *PR2*, *PR5*, and *PAL1*) and JA/ET-related genes (*PR3*, *PR4a*, *PR9*, and *PDF1.2*) in wheat seedlings upon *F. graminearum* infection. In summary, *S. pratensis* S10 could be an integrated biological agent and biofertilizer in wheat seedling blight management and plant productivity enhancement.

## 1. Introduction

*Fusarium graminearum*, a major pathogen causing Fusarium head blight (FHB) in wheat and other cereal crops, is regarded as one of the most devastating and prevalent fungal phytopathogens due to its economic importance and trichothecene toxicity [1,2]. *F. graminearum* can also result in seedling blight in major grain cereal crops, which damages seedlings and reduces plant establishment, consequently leading to substantial yield loss and crop quality reduction [3]. Currently, fungal disease management mainly relies on chemical fungicides (e.g., carbendazim and tebuconazole). Though the application of chemical fungicides efficiently inhibits *F. graminearum* infection, the indiscriminate use of fungicides has been confirmed to pose extensive adverse effects, such as increasing fungicide resistance, threating human and animal health, and causing serious environmental pollution [4]. Therefore, it is essential to develop environmentally friendly alternative strategies with low toxicity to effectively suppress *F. graminearum* infection.

Among the alternatives, biological agents (BCAs) for preventing phytopathogens are attracting widespread attention. *Streptomyces* spp. are the most abundant genus of actinobacteria with great potential to secrete multiple secondary metabolites [5,6,7]. Recently, *Streptomyces* spp. have been reported to efficiently inhibit a broad spectrum of fungal disease, e.g., Fusarium cotton wilt [8], wheat stripe rust [9], and tomato gray mold [10]. *Streptomyces* spp. mitigate pathogen infection in plants by different mechanisms, including the synthesis of antibiotics, mycoparasitism, physical displacement, competition, and the induction of plant systemic resistance [8,11]. Some *Streptomyces* species are plant growth-promoting streptomycetes (PGPSs) that have established favorable interactions with plants [12,13,14]. Generally, PGPSs promote plant growth via a variety of mechanisms, including nitrogen fixation, phytohormone generation, 1-aminocyclopropane-1-carboxylic acid (ACC) deaminase, and siderophore production, thereby alleviating abiotic or biotic stress to plants [8,15]. For instance, *S. hygroscopicus* TP-A0451 generates pteridic acid A and B with auxin-like activity, inducing adventitious root formation and stimulating root elongation in *Phaseolus vulgaris* hypocotyls [16]. *S. filipinensis* no. 15 reduces the endogenous level of ACC, consequently improving the resistance of tomato plants to abiotic and biotic stress [17]. Siderophore compounds are potential plant growth promoters and disease suppressors [18]. *Streptomyces* spp. were reported to promote their colonization in plants by producing siderophores, as well as assist in obtaining iron nutrition and limit and inhibit the growth of plant pathogens through iron competition [19].

Except for plant growth-promoting (PGP) properties, *Streptomyces* spp. have been demonstrated to induce resistance in plants to counteract pathogen infection [9]. Induced resistance (IR) has been investigated widely in numerous plant–pathogen interactions. Generally, IR is divided into induced systemic resistance (ISR) and systemic acquired resistance (SAR) [20]. ISR is a systemic resistance activated by nonpathogenic microbes that inhibit pathogen invasion [21]. Ethylene (ET) and jasmonate (JA) exert important modulatory roles in ISR [11]. Beneficial microbes activate ISR via triggering the plant for potentiated initiation of different cellular defense responses, such as ROS accumulation, callose deposition, defensed-related enzyme enhancement, and cell wall reinforcement [11,22]. In contrast, SAR is an induced resistance that occurs throughout the plants in response to a temporally earlier local exposure to a pathogen, which needs the biosynthesis of phenolic signaling compounds, such as salicylic acid (SA) [20]. Nevertheless, it was reported that antagonistic relationships occur between the JA/ET and SA pathways, and synergistic JA/ET and SA pathway interactions have also been detected. For example, *Trichoderma atroviride* IMI 206040, colonized on *Arabidopsis* roots, primes the expression of the JA/ET and SA pathways simultaneously to inhibit pathogen attack [23].

In our previous study, we observed that the application of S10 on wheat heads can effectively mitigate FHB incidence and deoxynivalenol (DON) content at anthesis [24]. However, strain S10 was investigated only for its direct action, and its indirect action mechanisms are not known. Thus, to better understand the indirect action mechanisms of strain S10 on wheat plants responses to *F. graminearum* stress, the objectives of this study were to (i) explore the plant growth-promoting traits of S10 in vitro; (ii) assess the growth-promoting activity of strain S10 in wheat plants; (iii) analyze the mechanisms of its growth-promoting activity based on genome mining; and (iv) evaluate the potential of S10 to induce defense resistance in wheat seedlings. These findings provide a potential solution for effectively controlling wheat seedling blight, contributing to sustainable agricultural development.

## 2. Materials and Methods

### 2.1. Strains and Culture Conditions

*Fusarium graminearum* strain PH-1 (NRRL 31084), originally isolated from corn in Michigan, was provided by the Mycology Laboratory of the Northwest A&F University (Xianyang, China). The strain was grown on potato dextrose agar (PDA) at 28 °C [25]. Five fresh mycelial blocks (6 mm diameter) were cut from the periphery of the colonies, added to 100 mL liquid sodium carboxymethyl cellulose (CMC) medium, and filtered through sterile cheesecloth to collect conidia. The suspensions were prepared by resuspending the conidia collected on the cheesecloth in sterile distilled water (SDW) and adjusting the concentration to 10^6^ conidia/mL using a hemocytometer. *Streptomyces pratensis* S10 (preserved at the Plant Disease Biological Control Laboratory of Northwest A&F University) was originally isolated from tomato leaf mold in Yangling, Shaanxi Province, and was grown on Mannitol soybean (MS) agar for 5 days at 28 °C. The strain fermentations were incubated at 28 °C, 180 rpm on a rotatory shaker (12 mm orbit) in liquid Gauze’s medium No. 1 (GS) for a week. The resulting suspension was filtered through sterile cheesecloth to obtain spores. The spore suspensions were prepared by resuspending the spores collected on the cheesecloth in SDW and adjusting the concentration to 10^8^ CFU/mL using a hemocytometer for use in the experiments.

### 2.2. Exploration of Plant Growth-Promoting Properties of S10 In Vitro

#### 2.2.1. Siderophore Production

Chrome-Azurol S (CAS) agar was applied to qualitatively determine the siderophore production by S10 [26,27]. Freshly grown S10 cultures were spot-inoculated on the CAS agar medium and cultured for 5 days at 28 °C. After incubation, the formation of a clear zone around S10 colonies suggests the production of siderophores. The experiment was conducted three times independently.

#### 2.2.2. Indole-3-acetic Acid (IAA) Production

The detection of IAA production was carried out by Salkowski’s colorimetric approach [28]. In brief, stain S10 was added to GS liquid medium and cultured at 28 °C, 180 rpm for a week. Then, 1 mL of cell-free supernatant was mixed with two milliliters of Salkowski reagent. After incubation for 30 min, the emergence of the red color means the generation of IAA. The non-inoculated cell-free suspensions served as a control. Three independent experiments were conducted, with three biological replicates per treatment.

#### 2.2.3. 1-Aminocyclopropane-1-carboxylate Deaminase (ACC) Detection

The ACC activity of S10 was detected by using Dworkin and Foster (DF) medium and ADF medium, containing ACC (in a final concentration of 3 mM) as the sole nitrogen source [29]. The above mediums were inoculated with S10 and cultured at 28 °C for a week. Strain S10 grew well in ADF medium and exhibited no growth on DF medium, indicating the generation of ACC deaminase. Three independent experiments were performed, with three biological replicates per treatment.

#### 2.2.4. Nitrogen Fixation Assay

The strain S10 was evaluated for nitrogen fixation ability on Burk’s modified N-free medium (pH 7.0) [30]. In brief, S10 colonies were cultured on nitrogen-free medium and cultured at 28 °C for a week. The appearance of strain S10 growth on the medium was regarded as a positive test for the nitrogen fixation experiment. Three independent experiments were conducted, with three biological replicates per treatment.

#### 2.2.5. Phosphate Dissolution

The insoluble organic phosphate-dissolving ability of S10 was qualitatively detected by the Pikovskaya (PVK) medium [26]. The agar plate was inoculated with S10 and cultured at 28 °C for a week. The formation of transparent halos around S10 colonies suggests positive phosphate solubilization. Three independent experiments were conducted, with three biological replicates per treatment.

#### 2.2.6. Hydrogen Cyanide Production

The capacity of S10 to produce hydrogen cyanide (HCN) was evaluated in GS medium amended with glycine (4.4 g/L) [31]. The agar plates were inoculated with S10 and incubated at 28 °C for a week. A change from yellow to red on the medium suggests HCN production. Each treatment had three biological replicates, and the experiment was performed three times, independently.

#### 2.2.7. Ammonia Production 

The fermentation broth of S10 was added to 1 mL of peptone water and cultured at 28 °C, 180 rpm for two weeks. Nessler’s reagent (0.5 mL) was then added into each tube, and the ammonia production was evaluated by the formation of a yellow-brown color [32]. The experiment was conducted three times independently, with three biological replicates per treatment.

### 2.3. Assessment of Plant Growth-Promoting Ability of S10

#### 2.3.1. Pot Experiment

To evaluate the plant growth promotion activity of S10 on wheat plants, a pot experiment was performed. Wheat seeds (Mingxian 169) were surface-disinfected through immersing in 75% ethanol for 2 min, soaked in 2% NaClO for 2 min, and washed five times in SDW. Subsequently, these disinfected seeds were sown in trays containing SDW for 7 days. Then, the same healthy seedlings were selected and cultivated in 10 cm diam pots (substrate was disinfected in autoclaves at 121 °C for 60 min). Three days after transplantation, wheat plants were inoculated with 30 mL of different concentrations of S10 spore suspension (10^7^, 10^8^, 10^9^ CFU/mL) with root drenching methods for each treatment. Wheat plants treated with 30 mL SDW were used as a control. Simultaneously, each pot was watered with 30 mL SDW once every 5 days. Then, 20 days after planting, the plants were gently taken out of the pots and rinsed with tap water to clear adhering particles. Biometric properties (dry weight, fresh weight, root length, and shoot length) were measured and statistically analyzed at the same time. Three independent experiments were conducted, and each treatment contained five replicate pots, with each replicate pot containing 10 wheat plants.

#### 2.3.2. Analysis of Root Activity

The triphenyl tetrazolium chloride (TTC) approach was applied to measure root activities [32]. Wheat plant root tips weighing 500 mg were immersed in a reaction solution consisting of 5 mL of phosphate buffer (PBS, pH 7.5) and 5 mL of 0.4% TTC. Then, the mixture was incubated at 37 °C for 1 h in the dark. The reaction was ended via the addition of 1 mol/mL, 2 mL of H_2_SO_4_. For the control, H_2_SO_4_ was added before the root tips, and then, the other procedures were conducted as described above. After the reaction was ended, the samples were transferred onto filter papers and cut into small pieces. Subsequently, the samples were ground with 5 mL of acetone and a small amount of quartz sand to extract triphenylformazan (TTF). Then, the red extract was transferred to a 10 mL volumetric flask and fixed with acetone. The extract absorbance was detected at 485 nm by a U-T3A spectrophotometer (Shanghai YiPu Instrument Manufacturing Co., Ltd., Shanghai, China), and a standard curve was created to detect TTF content based on the obtained absorbance value [33]. The root activity was expressed as the TTC reduction intensity according to the following equation: root activity = amount of TTC reduction (μg)/fresh root weight (g) × time (h). Three replicates were measured per treatment, and the experiment was carried out three times, independently.

#### 2.3.3. Chlorophyll Content

The chlorophyll level was measured following previous procedures [28]. Briefly, 100 mg of fresh wheat leaf sample was collected per treatment, homogenized in 80% acetone, and centrifuged for 5 min at 5000 rpm. Afterwards, the supernatant was collected, and the absorbance was examined by the U-T3A spectrophotometer at 663 nm and 645 nm. Three replicates were measured per treatment, and three independent experiments were performed.

### 2.4. Resistance-Inducing Assay

Seven-day-old wheat seedlings were selected as the experimental object. Then, 30 mL of the S10 spore suspension (1 × 10^8^ CFU/mL) was inoculated into the soil where the plants were grown for 24 h before the infection of *F. graminearum*. The conidial suspension of *F. graminearum* (10 μL, 1 × 10^6^ conidia/mL) was inoculated into artificial wounds on the wheat stem. All treated and control plants were then cultured in a growth chamber at 25 °C with a 12 photoperiod and 80% humidity. Wheat seedlings were harvested at 0, 12, 24, 48, 72, and 96 h after inoculation, immediately immersed in liquid nitrogen, and kept at −80 °C until they were subjected to extraction for enzyme and RNA extraction. Four treatments were set, including (1) wheat plants treated with SDW used as the control; (2) wheat plants inoculated with *F. graminearum* only (Fg); (3) wheat plants treated with S10 only (S10); and (4) wheat plants pretreated with S10 and then inoculated with *F. graminearum* (S10 + Fg). The disease index was classified according to the ratio of brown discoloration at the stem with a 0 to 4 scale reported previously [34]. A score of 0 represented no visible discoloration, and grades of 1, 3, 5, and 7 described discoloration of 0 to 25%, 26 to 50%, 51 to 75%, and 76 to 100%, respectively. The disease incidence was calculated by the following formula: the diseased plants/total number of investigated plants × 100%. The formula [(Σ number of diseased plants in each class × each evaluation class)/(total number of investigated plants × 7)] ×100 was used to calculate the disease index (DI) of each treatment. Three independent experiments were conducted, and each replicate had five replicate pots in each treatment, with each replicate pot containing 10 wheat plants.

### 2.5. Callose Deposition Detection

Twenty-four and forty-eight hours post inoculation (hpi), wheat seedlings were collected from different treatment groups and destained in 1:3 acetic acid/ethanol (*vol*/*vol*) to clear chlorophyll. Afterwards, the chlorophyll-free wheat stems were carefully washed with ddH_2_O. Then, the samples were immersed in 0.01% aniline blue staining solution containing K_2_HPO_4_ (150 mmol/L, pH 9.5) and incubated for 4 h in the dark [20]. The wheat stems were carefully washed with ddH_2_O, and the epidermis was detected by a fluorescence microscope (Olympus BX-53, Ishikawa-cho, Hachioji, Tokyo, Japan) with a UV excitation filter at 40× magnification. The level of callose deposition was quantified according to the fluorescent intensity by ImageJ software (version 1.8.0). Three replicates were examined per treatment, and three independent experiments were performed.

### 2.6. ROS Staining and Determination of H_2_O_2_

ROS accumulation was assessed by 3,3′-diaminobenzidine (DAB, Sigma, Shanghai, China) staining [20]. In brief, wheat stems from different treatment groups were stained with 1 mg/mL DAB solution (pH 3.8) at 25 °C overnight in the dark and cleared by destaining solution (acetic acid/ethanol = 1: 3, *vol*/*vol*) until the tissues were transparent. Then, the tissues were detected by an Olympus BX-53 microscope (Ishikawa-cho, Hachioji, Tokyo, Japan) at 40× magnification at 24 and 48 hpi. For the determination of H_2_O_2_, wheat seedlings were collected and ground with liquid nitrogen using a pestle and mortar. Then, the H_2_O_2_ levels of wheat seedlings were quantified by the Hydrogen Peroxide Content Assay Kit (BC3590, Solarbio, Beijing, China) following the manufacturer’s protocols. Three independent experiments were performed, and three biological replicates were examined per treatment.

### 2.7. Detection of Defense-Related Enzyme Activity

Wheat seedlings were sampled at 0, 12, 24, 48, 72, and 96 h for defense-related enzyme activity analysis. The samples (0.5 g) were ground in liquid nitrogen and mechanically homogenized in an ice bath in 5 mL PBS (pH 6.8, 20 mmol/L). The homogenate (crude enzyme extract) was centrifuged at 8000 rpm, 4 °C for 10 min. The obtained supernatant was kept at 4 °C for use in further determination. Superoxide dismutase (SOD) activity, catalase (CAT) activity, peroxidase (POD) activity, polyphenol oxidase (PPO) activity, and phenylalanine ammonia lyase (PAL) activity were detected using the Assay Kit (Solarbio, Beijing, China) following the manufacturer’s protocols. Three replicates were measured per treatment, and three independent experiments were conducted.

### 2.8. RNA Isolation and qRT-PCR

Quantitative real-time PCR (qRT-PCR) was applied to determine the expression levels of defense-related genes of wheat seedlings in different treatment groups. The wheat seedlings were collected at 0, 12, 24, 48, 72, and 96 h after inoculation with *F. graminearum*. The total RNA was extracted from the wheat tissues (50 mg) using a Quick RNA Isolation kit (Huayueyang Biotech, Beijing, China). The total RNA quality was evaluated with an Agilent 2100 bioanalyzer system (Agilent Technologies, Santa Clara, CA, USA) and a Nanodrop spectrophotometer 2000™ (Thermo Fisher Scientific, Shanghai, China). The first-strand cDNA was obtained with the FastKing cDNA Kit (Takara Biotech, Beijing, China). Then, qRT-PCR reactions were conducted by LightCycler apparatus (Roche Diagnostics GmbH, Switzerland) with the SYBR Green detection system. The relative *F. graminearum* biomass in each treatment was quantified by qPCR with the β-tubulin gene (FGSG_09530) as a reference to *F. graminearum* and wheat *Ta54227* primers (Appendix A) [23]. The expression of SA- (*PR1.1*, *PR2*, *PR5*, and *PAL1*) and JA/ET-related genes (*PR3*, *PR4a*, *PR9*, and plant defensin 1.2 *PDF1.2*) in wheat seedlings was assessed, and the *Ta54227* gene was applied as the internal reference. The relative expression levels of genes were analyzed with the 2^−ΔΔCt^ approach [8]. Each treatment had three replicates, and three independent experiments were conducted.

### 2.9. Statistical Analysis

All statistical analyses were conducted with IBM SPSS Statistics 26 (IBM Inc., Chicago, IL, USA) and GraphPad Prism 8 software (GraphPad Inc., San Diego, CA, USA). Values are shown as the mean ± standard error (SE). Multiple comparisons were analyzed by one-way analysis of variance (ANOVA) followed by Duncan’s multiple range tests. For each assay, the effect of the interaction experimental run × treatment was tested. As there was no difference between experimental runs, the data from the three runs of each bioassay were pooled. The differences between two groups were analyzed by the two-sided Student’s *t* test. All the experiments were conducted three times independently, and each treatment had three biological replicates.

## 3. Results

### 3.1. Plant Growth-Promoting Traits of S10 In Vitro

The development of a light red color by S10 on GS medium containing glycine indicates HCN production (Figure 1A). Distinct and clear hydrolytic halos were separately developed around the S10 colonies grown on solid medium containing CAS or Ca_3_(PO_4_)_2_ (Figure 1B,C), suggesting the production of siderophores and the decomposition of mineral phosphate. S10 showed no growth on the control DF medium, while it grew well on ADF medium, suggesting that S10 possesses the ability to secrete ACC deaminase (Figure 1D). Moreover, S10 showed the ability to decolor from brown to yellow in ammonia medium, revealing the production of ammonia (Figure 1E). Importantly, S10 was demonstrated to generate IAA through the red chromogenic reaction, and the IAA generation appeared to be high based on the red color (Figure 1F). S10 grew well on N-free Ashby medium, revealing that S10 possesses the potential to fix nitrogen (Figure 1G).

### 3.2. S. pratensis S10 Promoted Wheat Plant Growth

Based on the above results, the PGP effects of S10 were tested in wheat plants. As shown in Figure 2, the plant growth-promoting effect increased with the increased concentrations of S10 and then declined at a high concentration. The optimal effect was observed at a concentration of 10^8^ CFU/mL. When compared with the control, the quantitative results indicated that the shoot length, root length, fresh weight, dry weight, root activity, and chlorophyll content of the wheat seedlings inoculated with S10 spore suspension (at 10^8^ CFU/mL) were increased by 17.89, 23.08, 24.99, 26.98, 44.21, and 18.72%, respectively (Figure 2B–F).

### 3.3. Genome Mining of Genes Potentially Contributing to Plant Growth Promotion

To gain insights into the mechanisms of the growth promotion activities of S10, the genomic information of S10 (chromosome genome GenBank: CP051486; plasmid GenBank: CP051485) was analyzed.

#### 3.3.1. Plant Hormones

According to the genomic annotation, three trp-dependent IAA biosynthesis pathways, the tryptamine (TAM), indole-3-acetonitrile (IAN), and indole acetamide (IAM) pathways, appeared to participate in the IAA biosynthetic ability of S10 (Appendix A). In the IAM pathway, tryptophan can be directly transformed into IAM by a tryptophan monooxygenase enzyme. Finally, IAM is transformed into IAA through the action of the amidase enzyme. One tryptophan 2-monooxygenase encoding gene and four amidase encoding genes are detected in the S10 genome (Appendix A). In the TAM pathway, tryptophan can be transformed to TAM, followed by conversion to indole-3-acetaldehyde (IAAld) via an amine oxidase. Subsequently, the IAAld is transformed to IAA with the action of an aldehyde dehydrogenase enzyme. One monoamine oxidase and fifteen dehydrogenase encoding genes are found in the S10 genome (Appendix A). Additionally, S10 also processes a gene encoding nitrilase, participating in the process of converting IAN to IAA by the IAN pathway (Appendix A). Importantly, *trpA*, *trpB*, and *trpC* are, respectively, responsible for the tryptophan synthase alpha chain, the tryptophan synthase beta chain, and indole-3-glycerol phosphate synthase, which are related to tryptophan, synthesized by themselves. Some enzymes associated with tryptophan biosynthesis are also detected in the S10 chromosome (Appendix A). In addition, S10 also involves a putative ACC deaminase that can decompose ACC (the direct precursor of ethylene) (Appendix A).

#### 3.3.2. Siderophore Biosynthesis and Iron Absorption

The S10 genome possesses multiple genetic elements which participate in iron complex transport and siderophore synthesis (Appendix A). Notably, eight genes encoding the siderophore biosynthesis protein were found in the S10 genome. *entB*, *entE*, and *entH* participated in the production of enterobactin. Moreover, multiple genes are related to the biosynthesis of iron transport, such as the iron-siderophore ABC transporter substrate-binding protein, iron enterobactin transporter ATP-binding protein, and iron-siderophore uptake protein (Appendix A).

#### 3.3.3. Phosphate Solubilization

The S10 genome involves abundant genetic elements which participate in phosphate solubilization and transport (Appendix A). Noteworthily, strain S10 genomic DNA involves three *ppx-gppA* genes encoding exopolyphosphatase and a *ppa* gene encoding inorganic pyrophosphatase, which participate in the decomposition of inorganic polyphosphates. Sixteen genes encoding extracellular enzymes which participate in the solubilization of organic phosphate, such as alkaline phosphatase, polyphosphate glucokinase, and glyceraldehyde 3-phosphate reductase, were also found in the chromosome of S10 (Appendix A). Moreover, the *pstABCS* gene homologues associated with phosphate transport were predicted in the S10 genome and may enhance the ability of PO_4_ solubilization (Appendix A).

#### 3.3.4. Nitrate Reduction

Strain S10 harbors multiple nitrogen metabolism genes in the chromosome (Appendix A). The S10 genome contains *napA*, *napB, nirB*, and *nirD* genes encoding nitrate and nitrite reductase. Multiple genes encoding dissimilatory nitrate reduction, denitrification, and nitrification participate in the main nitrogen metabolism pathways.

### 3.4. S10 Induced Wheat Seedling Resistance against F. graminearum Infection

The results indicated that wheat seedlings only inoculated with *F. graminearum* showed obvious infection symptoms, with the development of dark brown stem lesions and blight (Figure 3A). In contrast, plants pretreated with S10 exhibited a significant reduction in disease symptoms, manifested by smaller blight size, compared with the Fg group. The disease incidence (*p* < 0.0001) and disease index (*p* < 0.0001) were reduced by nearly 52% and 65% in S10 + Fg-treated wheat seedlings compared to plants inoculated with *F. graminearum* only (Fg), respectively (Figure 3B,C). Consistently, the fungal biomass (*p* < 0.001) was substantially decreased in the S10 + Fg-treated wheat seedlings compared to those inoculated with *F. graminearum* only (Figure 3D). The fungal biomass decreased by 83.5% in the S10 + Fg group in comparison with the Fg group. The results indicated that S10 induced resistance in wheat seedlings against *F. graminearum*.

### 3.5. S10 Induced ROS Accumulation and Callose Deposition

The DAB staining results indicated that wheat seedlings inoculated with *F. graminearum* only (Fg) showed lighter coloration, whereas wheat seedlings in the S10 + Fg group exhibited darker staining at 24 hpi (Figure 4A). The S10 + Fg treatment group showed considerably more significant darker coloration than the Fg group at 48 hpi (Figure 4A), indicating that S10 induced ROS accumulation in wheat seedlings upon *F. graminearum* infection. Consistently, the H_2_O_2_ content in the S10 + Fg treatment group was 1.5- and 1.68-fold higher than in the Fg treatment group at 24 (*p* < 0.0001) and 48 hpi (*p* < 0.0001), respectively (Figure 4B). Callose deposition is ROS triggered defense for cell wall reinforcement. Callose deposition was noticeably enhanced in S10-treated together with *F. graminearum*-inoculated plants at 24 hpi and 48 hpi (Figure 4A). The S10 pretreatment resulted in more than 2-fold increases in the fluorescent intensities of callose deposition at 48 hpi (*p* < 0.0001), in comparison with the Fg treatment group (Figure 4C). Noteworthily, the S10 treatment group (S10) exhibited light ROS accumulation and callose deposition, indicating that they were triggered by *F. graminearum* infection instead of S10 treatment. Thus, S10 induced plant resistance upon pathogen attack and led to a rapidly activated cellular defense response.

### 3.6. S10 Enhanced Plant Defense Enzyme Activity

To further investigate whether S10 can increase the activity of the defense response of wheat plant cells, the defense-related enzyme activities in wheat seedlings were detected. The CAT activity showed a sharp increase in the S10 + Fg treatment, reaching the highest enzyme activity at 48 h, and then declined, at a level of 1.73-fold higher than that in the Fg group (Figure 5A). The SOD activity in the S10 + Fg group reached a peak at 12 h and subsequently declined. The highest enzyme activity was 1.23-fold higher than that in the Fg group (Figure 5B). In comparison with the Fg group, the POD activity in the S10 + Fg group increased to different levels after 0 h. Showing large changes, the POD activity reached its peak in the S10 + Fg group at 24 h, with an increase of 23.95%, compared with that in the Fg treatment group (Figure 5C). The PAL activity in the S10 + Fg group exhibited a rapid increase and reached the highest enzyme activity at 48 h, while that in the Fg group reached the peak at 72 h. The highest enzyme activity in the S10 + Fg group was 1.54-fold higher than that in the Fg group (Figure 5D). The PPO activity increased at a gradual rate in all treatment groups, reaching a peak at 48 h and 96 h in the S10 + Fg and Fg groups, respectively. The highest enzyme activity in the S10 + Fg group was 1.22-fold higher than that in the Fg group (Figure 5E). Noteworthily, all the activities of defense-related enzymes were maintained at higher levels in the S10 + Fg group than in the Fg group over the 96 h of the experiment.

### 3.7. S10 Induced the Expression of Defense-Related Genes in Wheat Seedlings

To evaluate whether S10 can induce the expression of defense-related genes in wheat seedlings, the expression levels of eight genes in wheat seedlings subjected to different treatments were determined. As presented in Figure 6, SA-related genes *PR1.1*, *PR2*, *PR5*, and *PAL1* in wheat plants inoculated with *F. graminearum* only displayed similar patterns. Compared with the Fg group, *PR1.1* and *PR2* all showed a maximum expression level at 24 hpi in the S10 + Fg-treated plants and increased by 3.67- and 5.18-fold, respectively (Figure 6A,B). The *PR5* expression in the S10 + Fg treatment group was obviously increased at 48 hpi and was 4.86-fold higher than that in the Fg group (Figure 6C). *PAL1* reached the highest expression level at 48 hpi in the S10 + Fg treatment group, with a fold change of 1.76, in comparison with the Fg group (Figure 6D). Additionally, with the JA/ET-related genes *PR3*, *PR4a*, and *PR9*, the overall dynamics of *PR3* expression were similar to those of *PR4a* expression (Figure 6E,F). The expression levels of the two genes in the S10 + Fg treatment group were all higher than those in the Fg treatment group. Remarkably, the relative expressions of genes *PR3* and *PR4a* both reached a maximum at 72 hpi, and the expression of *PR3* and *PR4a* in the S10 + Fg treatment group was 2.11- and 2.06-fold higher than in the Fg treatment group at this time (Figure 6E,F). In wheat seedlings treated with S10 + Fg, the gene *PR9* was obviously up-regulated at all time points in comparison with the Fg treatment group (Figure 6G). The *PR9* expression peaked at 48 hpi in the S10 + Fg treatment group, with 7.99-fold increases (Figure 6G). The expression of *PDF1.2* showed a slight up-regulation in the S10 + Fg treatment group, 2.25-fold higher in comparison with that in the Fg treatment group (Figure 6H). Overall, we found that when wheat seedlings were pretreated with S10, the JA/ET and SA signaling pathways were apparently activated in wheat plants but only upon *F. graminearum* infection. These results indicated that S10 modulated plant defense signaling pathways during pathogen infection.

## 4. Discussion

Plant growth-promoting biocontrol agents provide sustainable strategies for agriculture to adapt to abiotic and biotic stress. These agents modulate plant development and growth directly and indirectly. *Streptomyces pratensis* S10 inhibits fungal mycelial growth, conidiation, conidial germination, and perithecia production and reduces DON content in *F. graminearum* [24,35]. S10 produces a compound, venturicidin A, which significantly inhibits *F. graminearum* [36]. However, the effects of S10 on wheat plant growth and induced wheat plant resistance have not been investigated. In this study, we observed that root drenching with S10 alleviated the stress of *F. graminearum* in wheat seedlings by triggering the host defense mechanisms. More importantly, S10 greatly improved plant growth and development, contributing to alleviating pathogen stress in the plants.

Multiple BCAs possess the features of enhancing plant growth and promoting the acquisition and utilization of mineral nutrients, which contributes to disease inhibition [13,37]. In this study, S10 was confirmed to efficiently improve wheat plant growth in a growth chamber (Figure 2). The excellent efficiency of S10 in improving wheat plant development may result from the production of plant growth-promoting compounds and relevant genes in the genome (Appendix A). A previous report demonstrated that the production of IAA and ACC deaminase is extremely vital for the growth-promoting abilities of *Streptomyces* spp. [16]. S10 generated IAA or IAA-like compounds (Figure 1), which is consistent with the existence of three IAA biosynthesis pathways in its chromosome (Appendix A). IAA is a major character in plant growth-promoting BCAs [36], which improve the development of plant roots by boosting the elongation and proliferation of root cells [38]. S10 produces ACC deaminase in vitro (Figure 1) and harbors a gene related to the synthesis of ACC deaminase (Appendix A). ACC deaminase can decompose ACC (the direct precursor of ethylene), consequently suppressing ethylene biosynthesis in plants to enhance the capacity of plants to survive in unfavorable circumstances. Importantly, ACC deaminase was shown to synergistically interact with IAA and promote plant growth by regulating ethylene levels, one of the most important modulatory hormones of plant development [38,39].

S10 also displayed other plant growth promotion activities, comprising the production of HCN and siderophores, nitrogen fixation, and phosphate solubilization (Figure 1). Siderophore compounds facilitate plant nutrient absorption, acting as a direct mechanism in enhancing plant development, and they are also one of the most important mechanisms in inhibiting the plant pathogen by iron competition [8,40]. Furthermore, antiSMASH analysis found that S10 showed a prominent potential to biosynthesize several types of siderophores and a cluster involved in siderophores [24]. Siderophore compounds (catechol-type siderophores, hydroxamate-type siderophores, and mixed-type siderophores) produced by *Streptomyces* spp. can facilitate iron acquisition in the plant rhizosphere or prevent phytopathogens by reducing iron assimilation [41,42]. Phosphate deficiency is one of the restricted elements in crop yields. *S. griseus*-related strains with the ability to solubilize phosphate were demonstrated to alleviate the stress of *Pythium ultimum* in wheat [43].

*Streptomyces* spp. also can alleviate fungal disease stress in plants by inducing host plant resistance pathways. S10 was demonstrated to induce wheat seedling resistance to suppress *F. graminearum* infection (Figure 3). Previous studies have documented that *Streptomyces* spp. induce the plant defense response by the production of antioxidant enzymes, callose deposition, and ROS accumulation [8,10,30]. Callose deposition and ROS accumulation are hallmark events of plant early defense responses [44]. These physical and chemical defense mechanisms strengthen plant cell walls and enhance plant defense responses to invading pathogens [45]. Some BCAs have been shown to possess the ability to activate the defense response and safeguard plants from pathogen invasion through callose deposition [20]. For example, *Bacillus amyloliquefaciens* PMB05 induces callose deposition in strawberry leaves during *Colletotrichum gloeosporioides* infection [46]. *S. tauricus* XF primes ROS accumulation following the infection of wheat leaves by *Puccinia striiformis* f. sp. *tritici* [9]. Our results showed that S10 pretreatment enhanced callose deposition and ROS accumulation in wheat seedlings during *F. graminearum* infection (Figure 4). Together, these findings suggest that S10 can accelerate and improve the disease resistance ability of wheat seedlings through initiating the cellular defense response in systemic tissues.

Further results showed that S10 enhanced SOD, CAT, POD, PPO, and PAL activities to different degrees in wheat seedlings upon *F. graminearum* inoculation (Figure 5). Biocontrol agents activate the plant’s resistance to a pathogen generally in combination with the engagement of cell defense response activities, such as increased in defense-associated enzyme activities [11,33]. Different stress factors can result in the overaccumulation of ROS, consequently damaging the plant cells [47]. Appropriate ROS levels, nevertheless, could improve the tolerance to different types of stress [48]. For ROS to be applied as signaling molecules, they must be sustained at nontoxic levels via a delicate balance between ROS production and scavenging [49]. ROS-scavenging enzymes, comprising POD, CAT, and SOD, exert essential roles in the homeostasis of plant cellular ROS. In our results, S10 pretreatment greatly enhanced these ROS-scavenging enzymes activities in wheat seedlings during *F. graminearum* infection. The results indicated that S10 can remove excessive ROS in response to *F. graminearum* infection and enhance plant resistance to the invasion of *F. graminearum* through sustaining a balance in ROS levels. In addition, S10 also increased PPO and PAL activity. PAL participates in the synthesis of lignin and antibiotic compounds such as salicylic acid and flavonoids that limit the pathogen’s further development around the infection point [11]. PPO catalyzes the oxygen-dependent oxidation of phenols to quinones, consequently directly involved in disease resistance and the prevention of pathogen infection [16]. The systemic stimulation of PPO activities in response to pathogens can also provide another line of plant defense against the further invasion of pathogens [50]. These results demonstrated that S10 activated defense-related enzyme activities in wheat seedlings, alleviating *F. graminearum* stress.

A considerable number of studies reported that multiple *PR* gene expressions are induced upon pathogen attack and activate a series of protective responses in plants at a high level, thereby reducing or suppressing the growth and colonization of pathogens. A previous study indicated that *Bacillus cereus* AR156 triggered *PR* genes in *Arabidopsis* and tomato challenged with *Pseudomonas syringae* pv. *tomato* DC3000 [20]. Also, the colonization of soybean roots by *Klebsiella variicola* FH-1 induces the defense-related genes in soybean leaves against *Sclerotinia sclerotiorum* infection [31]. In this study, we found that the defense-related *PR* genes were highly expressed in S10-pretreated wheat seedlings challenged with *F. graminearum*, and they exhibited similar expression patterns (Figure 6). *PR1.1*, *PR2*, and *PR5* were up-regulated earlier after S10 + Fg treatment compared with the Fg treatment group. *PR1* is a target gene of the SA signaling pathway, which improves plant resistance to different pathogens [50,51]. The *PR2* gene has been identified as a typical regulator for SA and callose relied on in defense responses [11]. It encodes β-1,3-glucanase in the defense responses via decomposing cell walls for fungal pathogen attack. In addition, the *PR3*, *PR4*, and *PR9* genes are the JA/ET signaling pathway in plant defense, eventually preventing pathogen development [9]. Our results showed that the expression levels of the *PR3*, *PR4a*, and *PR9* genes were influenced by S10. Moreover, previous studies have confirmed that *PAL*, encoding for enzymes of the phenylpropanoid pathway, is expressed in challenges with different environmental stimuli, including pathogen attack and other biotic stress [52]. Consistent with the PAL enzyme activity results, S10 enhanced *PAL1* expression upon pathogen infection. *PDF1.2* is closely associated with the JA pathway, which is involved in the plant defense response to various stresses, including virus and fungal infection and adverse environments [53]. After pretreatment with S10, *PDF1.2* expression in wheat seedlings challenged with *F. graminearum* enhanced rapidly at 48 hpi (Figure 6). Taken together, the results suggest that S10 may induce wheat seedling resistance against *F. graminearum* via a synergistic activation between SA- and JA/ET-mediated pathways. Further research is needed to investigate the interaction mechanism between *S. pratensis* S10 and wheat plants to cope with *F. graminearum* infection.

## 5. Conclusions

In summary, S10 has the potential to be developed as a biofertilizer due to its capabilities in the production of IAA and siderophores, ACC deaminase activity, mineral phosphate solubilization, and nitrogen fixation. Notably, genome mining demonstrated that S10 processes abundant genes associated with plant growth regulation. In addition, S10 mitigated wheat seedling blight by inducing plant defense responses. S10 directly induced ROS accumulation and callose deposition in response to *F. graminearum* at the early infection stages. The increase in the activities of SOD, CAT, POD, PPO, and PAL may contribute to decreasing *F. graminearum* development. S10 also induced the expression of SA- (*PR1.1*, *PR2*, *PR5*, and *PAL1*) and JA/ET-related genes (*PR3*, *PR4a*, *PR9*, and *PDF1.2*) in wheat seedlings upon *F. graminearum* infection. This study indicates that S10 has great potential to be developed as a bioagent and as a biofertilizer.

## Figures and Tables

**Figure 1 jof-10-00578-f001:**
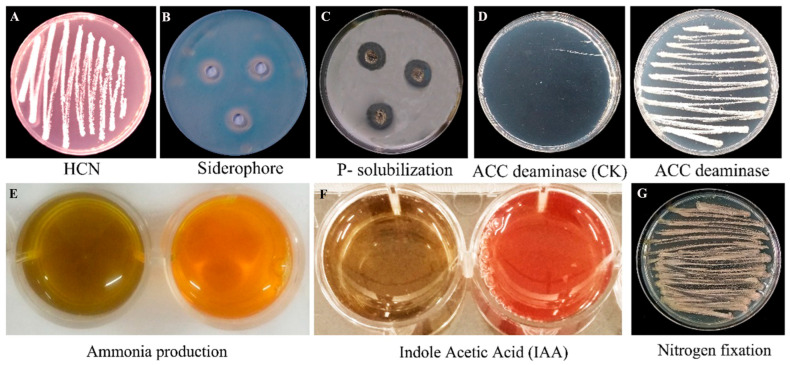
Evaluation of plant growth-promoting traits of *S. pratensis* S10 in vitro. (**A**) Hydrogen cyanide (HCN) production. (**B**) Siderophore production. (**C**) Phosphate solubilization activity. (**D**) 1-aminocyclopropane-1-carboxylic acid deaminase (ACC) activity. Left is control and right indicates ACC production. (**E**) Ammonia production. Left is control and right indicates ammonia production. (**F**) Indole-3-acetic acid (IAA) production. Left is control and right indicates IAA production. (**G**) Nitrogen fixation ability.

**Figure 2 jof-10-00578-f002:**
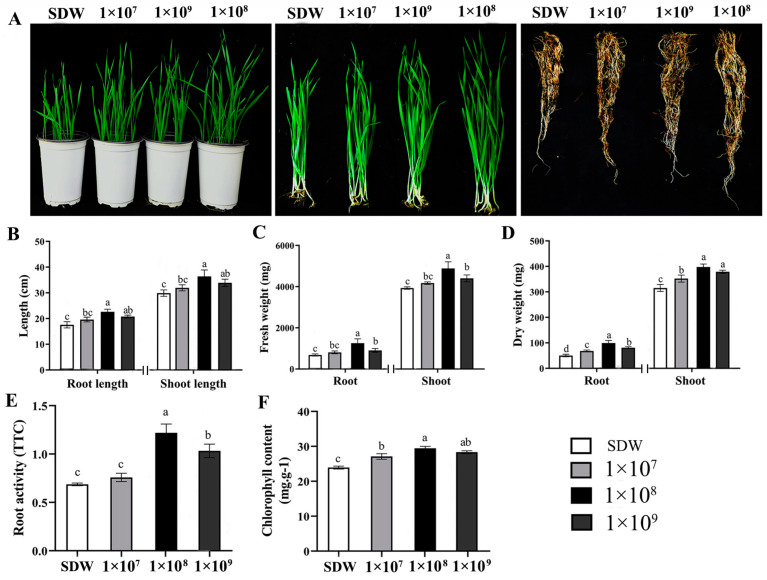
The wheat plant growth-promoting activity of *S. pratensis* S10. (**A**) Promoting effects of strain S10 on wheat growth and development in pot experiment. Overall appearance of wheat shoots and roots. Photos were captured at 20 days post inoculation (dpi) with different concentrations of S10 spore suspension. (**B**–**F**) Lengths, fresh weight, and dry weight of wheat roots and shoots as well as root activity (TTC) and chlorophyll contents of plants were measured at 20 dpi with different concentrations of S10 spore suspension. Data are presented as mean ± SE of three independent experiments consisting of three biological replicates (*n* = 50, *p* < 0.05, ANOVA, Duncan’s test). Lowercase letters on bars indicate significant differences. SDW represents wheat plants treated with sterile distilled water used as a control. 1 × 10^7^, 1 × 10^8^, and 1 × 10^9^ indicate the different concentrations of S10 spore suspension (CFU/mL).

**Figure 3 jof-10-00578-f003:**
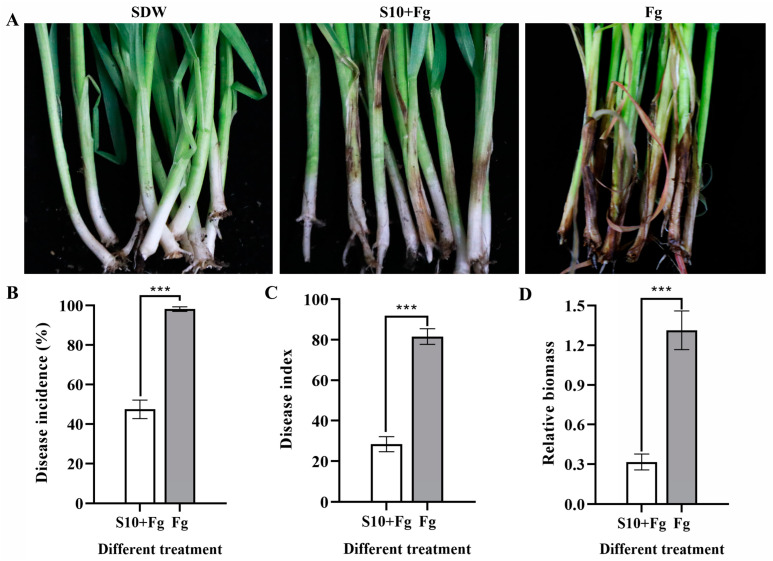
Inducing resistance in wheat seedlings against *F. graminearum* infection by *S. pratensis* S10. (**A**) Appearance of wheat seedlings changed with *F. graminearum* in absence/presence of *S. pratensis* S10. The photos were captured at 10 days post inoculation (dpi). SDW represents wheat plants treated with sterile distilled water used as a control; S10+ Fg means the wheat seedlings were pretreated with S10 24 h prior and then inoculated with *F. graminearum*; and Fg represents the wheat seedlings that were inoculated with *F. graminearum* only. (**B**) The disease incidence of wheat seedling blight was recorded at 10 dpi. Data are presented as mean ± SE of three independent experiments consisting of three biological replicates (Student’s *t* test). Asterisks indicate significant differences (***, *p* < 0.001). (**C**) The disease index of seedling blight was evaluated at 10 dpi. Data are presented as mean ± SE of three independent experiments consisting of three biological replicates (Student’s *t* test). Asterisks indicate significant differences (***, *p* < 0.001). (**D**) Relative biomass of *F. graminearum* in infected wheat seedlings was detected by qPCR. Mean ±SE were estimated with data from three independent experiments consisting of three biological replicates (Student’s *t* test). Asterisks indicate significant differences (***, *p* < 0.001).

**Figure 4 jof-10-00578-f004:**
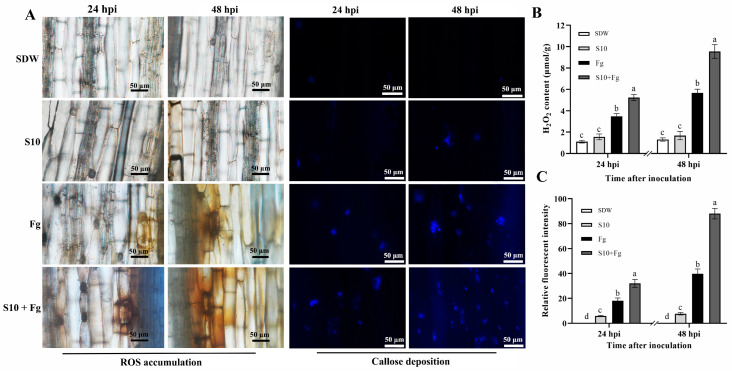
Effects of *S. pratensis S10* on ROS accumulation and callose deposition in wheat seedlings. (**A**) ROS accumulation and callose deposition in wheat seedlings in different treatment groups at 24 and 48 days post inoculation (dpi), respectively. Bars = 50 μm. (**B**) The H_2_O_2_ content in different treatment groups. Data are presented as mean ± SE of three independent experiments consisting of three biological replicates (*p* < 0.05, ANOVA, Duncan’s test). (**C**) The relative fluorescent intensity of callose deposition was evaluated by the ImageJ software. Data are presented as mean ± SE of three independent experiments consisting of three biological replicates (*p* < 0.05, ANOVA, Duncan’s test). Lowercase letters on bars indicate significant differences. SDW represents wheat plants treated with sterile distilled water used as a control; S10 means the wheat seedlings were treated with *S. pratensis* S10 only; Fg indicates the wheat seedlings that were inoculated with *F. graminearum* only; and S10 + Fg represents the wheat seedlings that were pretreated with S10 24 h prior and then inoculated with *F. graminearum*.

**Figure 5 jof-10-00578-f005:**
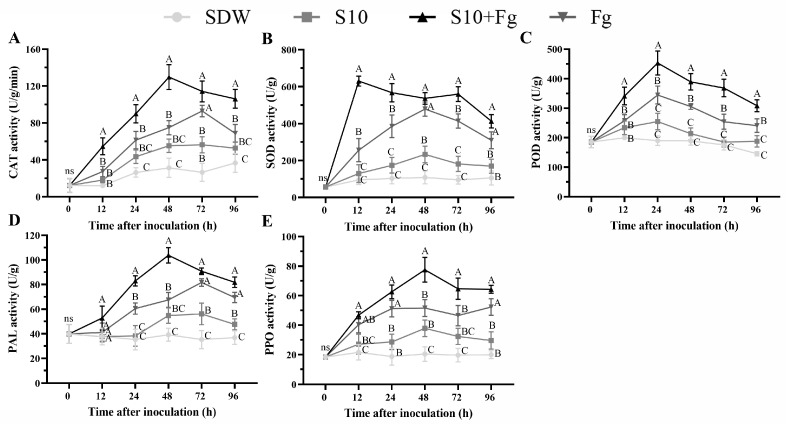
Effects of *S. pratensis* S10 on the activities of defense-related enzymes in wheat seedlings. (**A**) Catalase (CAT) activity. (**B**) Superoxide dismutase (SOD) activity. (**C**) Peroxidase (POD) activity. (**D**) Phenylalanineammonialyase (PAL) activity. (**E**) Polyphenoloxidase (PPO) activity. Data are presented as mean ± SE of three independent experiments consisting of three biological replicates (ANOVA, Duncan’s test). Capital letters (A, B, C) on bars indicate significant differences between different treatments at the same time, and ns indicates no statistical difference based on Duncan’s test (*p* < 0.01). SDW represents wheat plants treated with sterile distilled water used as a control; S10 means the wheat seedlings were treated with *S. pratensis* S10 only; Fg indicates the wheat seedlings that were inoculated with *F. graminearum* only; and S10 + Fg represents the wheat seedlings that were pretreated with S10 24 h prior and then inoculated with *F. graminearum*.

**Figure 6 jof-10-00578-f006:**
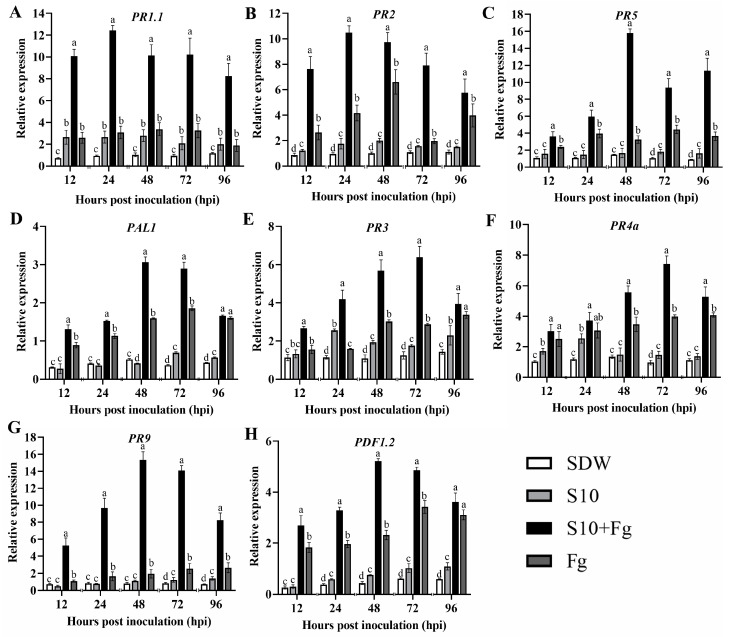
The expression levels of eight defense-related genes. The effects of *S. pratensis* S10 on PR1.1 (**A**), PR2 (**B**), PR5 (**C**), PAL1 (**D**), PR3 (**E**), PR4a (**F**), PR9 (**G**), and PDF1.2 (**H**) expression in wheat seedlings. Data are presented as mean ± SE of three independent experiments consisting of three biological replicates (*p* < 0.05, ANOVA, Duncan’s test). Lowercase letters on bars indicate significant differences. SDW represents wheat plants treated with sterile distilled water used as a control; S10 means the wheat seedlings were treated with *S. pratensis* S10 only; Fg indicates the wheat seedlings that were inoculated with *F. graminearum* only; and S10 + Fg represents the wheat seedlings that were pretreated with S10 24 h prior and then inoculated with *F. graminearum*.

## Data Availability

The original contributions presented in this study are included in this article; further inquiries can be directed to the corresponding author.

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
