# Peer review of "Streptomyces pratensis S10 Promotes Wheat Plant Growth and Induces Resistance in Wheat Seedlings against Fusarium graminearum"

_jof, 2024, doi:10.3390/jof10080578_

Round 1
Reviewer 1 Report
The present work demonstrates updated and improved methods on the topic.
It develops its work in detail, although some adjustments in the description of materials and methods should be reviewed. The subject studied in wheat should be reviewed by authors who have experience on the subject.I congratulate the authors for the work done.
Consider writing more about siderophores in introduction or explaining the essay
Line 117: dice stain
123 there is a or that does not understand. the meaning of the sentences.
125 “strain” is missing
Line 126: Name ADF medium correctly
22.3: ACC Detection needs more explanation as how is this experiment working on S10 strain growing.
2.2.4: need to be improved.
22.5: Ok, is ther any chance that you can make small description of the medium you are using in each assay or it is enough for the editorial manager?
Line 218: what is ms?
Please check language
Author Response
Dear Reviewer,
Thank you very much for your comments and professional advice. These comments help us to improve the manuscript. We have studied comments carefully point by point and have made correction and modification. We hope the manuscript can be improved this time.
I hope the information is clearer this time and if there is any problem, please help us to point it out. Thanks, and I wish you all the best.
Paragraphs below are our point-by-point responses (in red color).
The present work demonstrates updated and improved methods on the topic.
It develops its work in detail, although some adjustments in the description of materials and methods should be reviewed. The subject studied in wheat should be reviewed by authors who have experience on the subject.
I congratulate the authors for the work done.
Response: Thank you for your recognition. We hope this revised manuscript is better.
Comments 1: Consider writing more about siderophores in introduction or explaining the essay.
Response 1: Thank you for pointing this out. We agree with this comment. We have written more about siderophores in introduction (Line 62-65).
Comments 2: Line 117: dice stain
Response 2: Thank you for pointing this out. Yes, dice strain.
Comments 3: 123 there is a or that does not understand. the meaning of the sentences.
Response 3: Thank you for pointing this out. We have changed into "The ACC activity of S10 was detected by using Dworkin and Foster (DF) medium and ADF medium (ammonium sulfate as a sole nitrogen resource)"(Line 126-127). Thank you again.
Comments 4: 125 “strain” is missing
Response 4: Thank you for pointing this out. We added the "strain" (Line 129).
Comments 5: Line 126: Name ADF medium correctly
Response 5: Thank you for pointing this out. ADF medium name is correctly. S10 strain grew well in ADF medium and exhibited no growth on DF medium, indicating the generation of ACC deaminase. The DF medium is a control medium (Line 129).
Comments 6: 2.2.3: ACC Detection needs more explanation as how is this experiment working on S10 strain growing.
Response 6: Thank you for pointing this out. We added the explanation as how is this experiment working on S10 strain growing (Line 129-130).
Comments 7: 2.2.4: need to be improved.
Response 7: Thank you for pointing this out. We have improved the section 2.2.4. Hope it's better this time.
Comments 8: 2.2.5: Ok, is ther any chance that you can make small description of the medium you are using in each assay or it is enough for the editorial manager?
Response 8: Thanks for pointing out. Yes, it is enough for the editorial manager.
Comments 9: Line 218: what is ms?
Response 9: Thanks for pointing out. Sorry for my obvious mistakes. We have change into "chlorophyll-free wheat stems " (Line 221). Thank you again.
Reviewer 2 Report
Dear authors
The ability of some microorganisms to regulate plant growth and suppress infections has been known for several decades. However, the mechanisms that ensure this are still being investigated. The authors of the manuscript investigated a promising bacterial strain Streptomyces pratensis S10 by subjecting it to an almost complete set of tests currently used for PGP bacteria. Overall, the study made a good impression. The choice and design of the experiments seem well thought out. Many different characteristics of the bacterial strain have been studied, as well as its effect on wheat plants at the biochemical level. Some of the phenotypic characteristics are confirmed by the presence of corresponding genes or changes in gene activity. Despite the variety of data, they are described in a logical sequence. The significance of each indicator for the beneficial interaction of plants and bacteria is explained. The results are well illustrated. The conclusions are based on a statistical analysis of the data. References include many articles in highly rated journals.
However, the conclusions lack accuracy due to the use of words like «defense-related». I recommend using the names of enzymes or groups of enzymes, the names of metabolic pathways to which genes relate, etc.
I have two small remarks about the conclusions.
1. It seems there is a dot missing after the word «responses» in line 595.
2. Line 596. «The increase in activities of defense-related enzymes indicated that S10 restricted F. graminearum development». The phrase seems unfortunate. After all, a direct relation between antioxidant enzymes and the development of the fungus has not been shown.
This also applies to the phrase line 590 « S10 evidently improved plant development and growth via a wide variety of mechanisms, including IAA biosynthesis, siderophore production, ACC deaminase activity, mineral phosphate solubilization, and nitrogen fixation». The bacterium has all the listed properties. But whether all of them contributed to the stimulation of wheat growth in your experiment is unknown.
Author Response
Dear Reviewer,
Thank you very much for your comments and professional advice. These comments help us to improve the manuscript. We have studied comments carefully point by point and have made correction and modification. We hope the manuscript can be improved this time.
I hope the information is clearer this time and if there is any problem, please help us to point it out. Thanks, and I wish you all the best.
The ability of some microorganisms to regulate plant growth and suppress infections has been known for several decades. However, the mechanisms that ensure this are still being investigated. The authors of the manuscript investigated a promising bacterial strain Streptomyces pratensis S10 by subjecting it to an almost complete set of tests currently used for PGP bacteria. Overall, the study made a good impression. The choice and design of the experiments seem well thought out. Many different characteristics of the bacterial strain have been studied, as well as its effect on wheat plants at the biochemical level. Some of the phenotypic characteristics are confirmed by the presence of corresponding genes or changes in gene activity. Despite the variety of data, they are described in a logical sequence. The significance of each indicator for the beneficial interaction of plants and bacteria is explained. The results are well illustrated. The conclusions are based on a statistical analysis of the data. References include many articles in highly rated journals.
However, the conclusions lack accuracy due to the use of words like «defense-related». I recommend using the names of enzymes or groups of enzymes, the names of metabolic pathways to which genes relate, etc.
Response: Thank you for your recognition. We added the names of enzymes in the conclusions.
Comments 1: It seems there is a dot missing after the word «responses» in line 595.
Response 1: Sorry for my obvious mistakes. Thanks for pointing out. We have added a dot after the word «responses» in line 597.
Comments 2: Line 596. «The increase in activities of defense-related enzymes indicated that S10 restricted F. graminearum development». The phrase seems unfortunate. After all, a direct relation between antioxidant enzymes and the development of the fungus has not been shown.
Response 2: Thanks for pointing out. We changed into “The increase in activities of SOD, CAT, POD, PPO, and PAL may contribute to decreasing F. graminearum development.” (Line 599-600). Hope it’s better this time.
Comments 3: This also applies to the phrase line 590 « S10 evidently improved plant development and growth via a wide variety of mechanisms, including IAA biosynthesis, siderophore production, ACC deaminase activity, mineral phosphate solubilization, and nitrogen fixation». The bacterium has all the listed properties. But whether all of them contributed to the stimulation of wheat growth in your experiment is unknown.
Response 3: Thanks for pointing out. We agree with your comments. So, we changed into “it has potential to be developed as bio-fertilizers due to their capabilities in the production of IAA, siderophore, ACC deaminase activity, mineral phosphate solubilization, and nitrogen fixation.” (Line 593-595).
Reviewer 3 Report
This paper appeared to have conducted the experiments honestly and evaluated the results sincerely. This is a new method of countering plant diseases and can be said to be a groundbreaking method that also promotes plant growth. A few corrections would likely result in a better paper.
Line 43: Please provide references regarding the adverse effects of indiscriminate use of chemical fungicides.
Figure 2B-F: Arrange the horizontal axis in ascending order of concentration (SDW, 1×107, 1×108, 1×109).
Author Response
Dear Reviewer,
Thank you very much for your comments and professional advice. These comments help us to improve the manuscript. We have studied comments carefully point by point and have made correction and modification. We hope the manuscript can be improved this time.
I hope the information is clearer this time and if there is any problem, please help us to point it out. Thanks, and I wish you all the best.
Paragraphs below are our point-by-point responses (in red color).
This paper appeared to have conducted the experiments honestly and evaluated the results sincerely. This is a new method of countering plant diseases and can be said to be a groundbreaking method that also promotes plant growth. A few corrections would likely result in a better paper.
Response: Thank you for your recognition. We hope this revised manuscript is better.
Comments 1: Line 43: Please provide references regarding the adverse effects of indiscriminate use of chemical fungicides.
Response 1: Thanks for pointing out. We provided references regarding the adverse effects of indiscriminate use of chemical fungicides.
Comments 2: Figure 2B-F: Arrange the horizontal axis in ascending order of concentration (SDW, 1×107, 1×108, 1×109).
Response 2: Thanks for pointing out. We arranged the horizontal axis in ascending order of concentration (SDW, 1×107, 1×108, 1×109).